# Angiopoietin-like 2 Protein and Hidradenitis Suppurativa: A New Biomarker for Disease Severity

**DOI:** 10.3390/biomedicines11041204

**Published:** 2023-04-18

**Authors:** José L. Hernández, J. Gonzalo Ocejo-Vinyals, Mónica Renuncio-García, Elena González-López, Ricardo Blanco, Marcos A. González-López

**Affiliations:** 1Internal Medicine Division, Hospital Universitario Marqués de Valdecilla, 39008 Santander, Spain; 2Medicine and Psychiatry Department, Universidad de Cantabria, 39011 Santander, Spain; 3Valdecilla Research Institute, Valdecilla (IDIVAL), 39011 Santander, Spain; 4Immunology Division, Hospital Universitario Marqués de Valdecilla, 39008 Santander, Spain; 5Rheumatology Division, Hospital Universitario Marqués de Valdecilla, 39008 Santander, Spain; 6Dermatology Division, Hospital Universitario Marqués de Valdecilla, 39008 Santander, Spain

**Keywords:** hidradenitis suppurativa, angiopoietin-like 2 protein, insulin resistance

## Abstract

Hidradenitis suppurativa (HS) is a chronic inflammatory disease whose pathogenesis is not fully understood at present. The role of proinflammatory cytokines, several adipokines, retinol-binding protein 4, angiopoietin-2 and other molecules has been previously reported. Angiopoietin-like 2 protein (ANGPTL2) is a glycoprotein belonging to the angiopoietin-like family that may play a pivotal role in the pathogenesis of several chronic inflammatory diseases. To our knowledge, the role of serum ANGPTL2 levels in HS has not been assessed to date. In the current case–control study, we aimed to investigate serum ANGPTL2 levels in HS patients and controls and to assess whether ANGPTL2 levels could be associated with the severity of HS. Ninety-four patients with HS and sixty controls of similar age and sex were included in the study. Demographic, anthropometric, and clinical data, as well as routine laboratory parameters and serum concentrations of ANGPTL2, were assessed in all participants. HS patients had significantly higher serum ANGPTL2 levels than controls after adjusting for confounders. Moreover, ANGPTL2 concentrations positively correlated with disease duration and severity. Our results indicate for the first time that serum ANGPTL2 concentrations are elevated in HS patients compared to controls and correlate with the duration of the disease. Besides, ANGPTL2 might serve as a biomarker of HS severity.

## 1. Introduction

Hidradenitis suppurativa (HS) is a recurrent painful and disabling chronic inflammatory disorder characterized by nodular lesions, abscesses, and fistulae most commonly in the apocrine gland-bearing areas of the body [1]. Its pathogenesis is multifactorial and not fully understood yet, which implies, in many cases, an unpredictable response to therapy.

The cytokines and the immune pathways involved in the development of HS have not been fully elucidated, but the upregulation of inflammatory cytokines, such as tumor necrosis factor-alpha (TNF-α), interleukin (IL)-17, IL-6, and IL-23 has been reported as a key pathophysiological factor for the disease [2,3]. Moreover, it has been described that high levels of TNF-α in skin lesions as well as IL-17 in the serum of HS patients correlate with disease severity, together with an inflammatory profile with a predominance of Th1 and Th17 lymphocytes [4].

Recently, several molecules and inflammatory markers, such as retinol-binding protein 4 (RBP4), angiopoietin-2 (Ang-2), neutrophil-lymphocyte ratio (NLR), and pan-immune-inflammation value (PIV), were identified as biomarkers of disease severity in patients with HS [5,6,7,8].

Angiopoietin-like-2 protein (ANGPTL2) is a glycoprotein belonging to the angiopoietin-like (ANGPTL) family, with an N-terminal coiled-coil domain, a short linker peptide, and a C-terminal fibrinogen-like domain [9].

It is highly expressed in adipose tissues and may play a pivotal role in some inflammatory processes such as obesity-related IR [10] and chronic systemic inflammatory diseases such as rheumatoid arthritis [11] or dermatomyositis [12].

ANGPTL2 signaling functions in angiogenesis and tissue repair, where an excess of ANGPTL2 signaling has been associated with chronic inflammation and subsequent pathological irreversible tissue remodeling. Overexpression of ANGPTL2 in skin tissue leads to focal inflammation and increases blood vessel permeability due to vascular inflammation [13]. Additionally, this protein has been linked to smoking, obesity, and insulin resistance (IR), three key factors in HS pathogenesis [10]. Thus, adipocyte-derived inflammatory ANGPTL2 has been proposed to link obesity to IR. In patients with diabetes and obesity, ANGPTL2 produced by adipocytes, infiltrated macrophages, and endothelial cells, leading to an inflammatory response through activation of the integrin α5β1/Rac1 NFκB pathway resulting in inflammatory gene expression [14]. In the Hisayama study [15], raised serum ANGPTL2 levels were positively associated with the development of type 2 diabetes mellitus in community-dwelling Japanese subjects.

High levels of ANGPTL2 have been proposed as a biomarker of chronic inflammatory disorders and early diagnosis, prognosis, and recurrences of some types of cancers. The link between this increase in serum ANGPLT2 levels and these chronic conditions is systemic inflammation and some other properties of this protein indirectly related to inflammation, such as its potential to contribute to cellular senescence [16].

Moreover, since vascular injury along with vascular inflammation are considered an early manifestation of arteriosclerosis, serum ANGPTL2 levels might be involved in the arteriosclerotic process and could act as a new biomarker of atherosclerosis [17]. Thus, ANGPTL2 is abundantly expressed in endothelial cells and macrophages infiltrating atheromatous plaques [18]. In the clinical setting, serum ANGPTL2 levels are highly expressed in patients with acute myocardial infarction and have been positively related to the severity of the coronary lesion [19].

In contrast to angiopoietins (Ang), ANGPTLs, despite sharing a great similarity in their amino acid and structural sequences [9], none of these latter bind Tie-1 or Tie-2, the endothelial-specific receptor tyrosine kinase 1 and 2 [20,21].

The binding of ANGPTL2 to integrin α5β1 enhances cell motility and extracellular matrix remodeling, leading to tissue repair [12]. In the same way, it can bind to type 1A angiotensin II receptors in the cytosol of several cells. Similarly, it binds to CD146, which is present in endothelial cells, preadipocytes, mature adipocytes, and other cells. Furthermore, it binds human leukocyte immunoglobulin-like receptor B2, also known as CD85 or immunoglobulin-like transcripts in bone marrow [22,23].

Several studies have focused on the role of ANGPTLs in certain conditions and disorders. Thus, ANGPTL2 signaling has been reported to be important in the atherosclerosis process, angiogenesis, chronic inflammation, metabolic and lipid disorders and some types of cancer [10,11,12,15,16].

To our knowledge, the role of serum ANGPTL2 levels in HS has not been assessed to date. Given the above, and based on our previous work regarding the role of several molecules related to endothelial dysfunction, atherosclerosis, and disease severity in patients with chronic inflammatory disorders, Ang-2 among others [6], we aimed to investigate whether there could be differences in serum ANGPTL2 levels in HS patients compared with healthy controls. Furthermore, we sought to assess whether there was any relationship between these levels and HS severity.

## 2. Materials and Methods

### 2.1. Participants and Protocol

This is a case–control study of 94 patients with HS and 60 controls, recruited from our dermatology outpatient clinic at a tertiary-care hospital in Santander, northern Spain, which serves as a reference center for a population of 350,000 inhabitants. HS patients were ≥18 years and all fulfilled the diagnostic criteria for HS. The diagnosis of HS is a clinical based diagnosis with no pathognomonic tests [1,24]. The control group was set up among subjects with non-inflammatory cutaneous disorders and hospital clinical staff of similar age and sex than patients. The study protocol was previously published [6]. Briefly, patients or controls were excluded for the following reasons: a personal history of CV events, diabetes mellitus, chronic kidney or liver failure, and/or other inflammatory diseases. The International Hidradenitis Suppurativa Severity Score System (IHS4) was used to assess HS severity (IHS4  ≤3 mild, 4–10 moderate, and ≥11 severe) [25]. Demographic, clinical, and physical examination data (height, weight, body mass index -BMI-, and blood pressure) were collected from all the participants according to a prespecified protocol. Metabolic syndrome was diagnosed according to the criteria proposed by the National Cholesterol Education Program’s Adult Treatment Panel III (ATP III). HS duration was collected in all the cases.

Blood samples were obtained from an antecubital vein in the morning after a requested 12 h overnight fast. Serum glucose, insulin, glycated hemoglobin (HbA1c), triglycerides, total serum cholesterol (TC), high-density lipoprotein cholesterol (HDL-c), low-density lipoprotein cholesterol (LDL-c), and high-sensitivity C-reactive protein (hs-CRP) levels were determined. These baseline parameters were measured using standard methods (ADVIA 2400 Chemistry System autoanalyzer; Siemens, Munich, Germany). The Homeostatic Model Assessment for IR (HOMA-IR), expressed as fasting insulin level (μIU/mL) × fasting glucose level (mg/dL)/405, was calculated to assess the degree of IR. IR was diagnosed when the HOMA-IR value was >2.5.

Serum ANGPTL2 levels were measured by an enzyme-linked immunosorbent assay according to the manufacturer’s instructions (Human ANGPTL2 ELISA kit, Cusabio, bioNovacientífica S.L., Barcelona, Spain). The sensitivity for ANGPTL2 was less than 0.39 ng/mL, and the intra-assay and the interassay coefficients of variation were <8.9% and <10%, respectively. The detection range was 1.56–100 ng/mL.

This study was approved by the Clinical Research Ethics Committee of Cantabria (internal code: 2013.267), and all participants gave written informed consent.

### 2.2. Statistical Analysis

All continuous variables were tested for normality. Results were expressed as numbers (percentage), mean ± standard deviation (SD), or median and interquartile range (IQR), as appropriate. Spearman’s rank correlation coefficients were calculated to assess the relationship between serum ANGPTL2 levels and demographic and laboratory parameters. Student’s *t*-test or the Mann–Whitney *U* test was used to determine the differences between groups for continuous variables and the χ^2^-test for categorical variables. Kruskal–Wallis test with Bonferroni correction was used to compare serum ANGPTL2 levels across the three IHS4 groups. Besides, general linear models, adjusted for age, sex, BMI, serum hs-CRP levels, and current smoking, were built to assess the association of ANGPTL2 with both HS and HS severity, according to the IHS4 groups (mild, moderate, and severe HS).

Statistical analyses were performed using IBM SPSS 28.0 (Armonk, NY, USA: IBM Corp) and Graph Pad Prism software 6.0. All *p*-values were two-tailed and a result <0.05 were considered to be statistically significant in all the calculations.

## 3. Results

This section may be divided into subheadings. It should provide a concise and precise description of the experimental results, their interpretation, as well as the experimental conclusions that can be drawn.

### 3.1. Baseline Features and Serum ANGPTL2

Table 1 shows the baseline demographic and clinical features and laboratory parameters of HS patients and controls. Noteworthy, HS patients had higher BMI, waist perimeter, and prevalence of smoking and IR than controls. They also had higher serum insulin, hs-CRP, and fibrinogen levels than control subjects. Twenty-four HS patients were on active tumor necrosis factor (TNF)-α inhibitors.

In HS patients, serum ANGPTL2 levels were related to age (rho = 0.304; *p* = 0.003), serum hs-CRP concentrations (rho = 0.269; *p* = 0.009), fibrinogen levels (rho = 0.342; *p* = 0.003), and duration of HS (rho = 0.242; *p* = 0.019). Additionally, there was a trend for an association between ANGPTL2 and serum insulin levels (rho = 0.192; *p* = 0.083) and IR (rho = 0.204; *p* = 0.066).

Median serum ANGPTL2 levels in HS patients and controls were 45.10 [30.24–57.97] and 0.37 [0.09–0.92] ng/mL, respectively (*p* < 0.0001) (Figure 1). These differences remained significant after adjusting for age, sex, BMI, serum hs-CRP levels, and current smoking (*p* < 0.0001).

### 3.2. ANGPTL2 and HS Severity

Serum ANGPTL2 levels were significantly associated with HS severity, measured by the IHS4 with the highest values in the most severe HS group (Figure 2). These differences remained significant after adjusting for age, sex, BMI, serum hs-CRP levels, and current tobacco use (*p* < 0.0001). Further adjustment for the duration of HS, metabolic syndrome, IR, serum fibrinogen levels, and the use of TNF-α inhibitors did not change these results.

## 4. Discussion

We have found, for the first time, that serum ANGPTL2 levels were significantly higher in HS patients than in controls of similar age and sex. Moreover, ANGPTL2 was related to HS severity measured by the IHS4, and this association is independent of age, BMI, metabolic syndrome, serum hs-CRP levels, and IR.

ANGPTL2 is a glycoprotein belonging to the angiopoietin-like (ANGPTL) family, with an N-terminal coiled-coil domain, a short linker peptide, and a C-terminal fibrinogen-like domain. It is highly expressed in adipose tissues and may play a pivotal role in some inflammatory processes such as obesity-related IR [14] and chronic systemic inflammatory diseases such as rheumatoid arthritis [12] or dermatomyositis [13].

Unlike other ANGPTLs, ANGPTL2 has the unique capacity to induce an inflammatory response in the blood vessels. It also promotes local inflammation in adipose tissue and systemic IR. Besides, this protein activates migration and inflammatory changes of endothelial cells and monocyte/macrophages via integrins. Additionally, constitutive activation of ANGPTL2 leads to local inflammation in mouse skin tissue. Thus, ANGPTL2 plays a key role in the mechanism underlying adipose tissue inflammation, which is involved in the pathogenesis of the IR associated with obesity [14].

Furthermore, experimental studies in mice have found that endothelial cell-derived ANGPTL2 accelerates vascular inflammation most likely by activating integrin α5β1/Rac1/NF-ΚB proinflammatory pathway in endothelial cells and increasing macrophage infiltration, leading to endothelial dysfunction and progression of the atherosclerotic process [26].

Serum ANGPTL2 levels are upregulated mainly in visceral obesity and related metabolic disorders. Besides, circulating ANGPTL2 concentrations have been correlated with BMI, waist perimeter, and serum CRP in healthy volunteers [9]. Because obesity, IR, and inflammation have been involved as key pathogenetic factors of HS, we hypothesized that raised serum ANGPTL2 levels could be related to HS severity.

By contrast, Ang-2 is a multifaceted protein expressed by vascular endothelium that acts as an antagonist of Angiopoietin-1 and the Tie2 receptor [27]. The Ang/Tie pathway acts as a regulator of angiogenesis under physiological and pathological conditions [28]. Ang-1 and Ang-2 bind to the endothelial-specific receptor tyrosine kinase (Tie2) with similar affinities but with different effects. Ang-1 induces Tie2 phosphorylation while Ang-2 which is the natural antagonist of Ang-1, blocks Tie2 phosphorylation [20]

Serum Ang-2 has been involved in endothelial activation, angiogenesis, atherosclerosis, and inflammatory mechanisms [29]. As there has been observed in other chronic inflammatory diseases, we have previously reported that serum Ang-2 levels were significantly higher in patients with HS compared to controls [6].

Furthermore, Ang-2 was positively correlated with disease severity in HS patients and other dermatological diseases, such as psoriasis [11], and atopic dermatitis [12]. Since the correlation between Ang-2 and CV disease is well known, the results of our study might indicate that high levels of Ang-2 could play a role in the CV burden of HS.

HS is now recognized as a systemic disease. The precise inflammatory mechanism underlying HS pathogenesis is still not fully elucidated, but elevated levels of proinflammatory cytokines and adhesion molecules, such as TNFα, IL-6, IL-17, and IL-1β have been observed in serum and HS skin lesions [2,3,30,31]. In this sense, ANGPTL2 may induce the expression of proinflammatory cytokines, such as TNFα, IL-6, and some adhesion molecules, that have been involved in the pathogenesis of HS and may contribute to maintaining a chronic inflammatory status characteristic of this disorder (Figure 3).

Besides, TNFα regulates ANGPTL2 mRNA expression and could potentiate the overall inflammatory effect [32]. In mice, it has been reported that ANGPTL2 exacerbates proinflammatory signaling by activating the Integrin α5β1/NF-κB pathway in coronary endothelial cells [26]. The nuclear factor-κB (NF-κB) is a transcription factor that develops a pivotal role in activating both innate and adaptive immunity, mainly leading to target cells to increase production of proinflammatory cytokines [33], such as IL-1β, TNFα and IL-6. The activation via receptors such as integrin α5β1 and leukocyte immunoglobulin-like receptor B2 would trigger a transduction signaling cascade increasing expression of inflammatory factors genes through NF-κB and other transcription factors, as has been demonstrated in synovial tissue [34]. NF-κB pathway activation has been associated with several inflammatory diseases, such as rheumatoid arthritis, inflammatory bowel disease, multiple sclerosis, and asthma [33].

In skin tissues, overexpression of ANGPTL2 leads to local inflammation and increased vascular permeability through vascular inflammation but not angiogenesis, since blood vessels seem to remain unaltered [13]. In addition to adipocytes, several cells involved in psoriasis and HS pathogenesis can secrete ANGPTL2, such as macrophages, keratinocytes, and endothelial cells. In a recent report, Kenawy et al. [35], found that serum ANGPTL2 levels were elevated in patients with psoriasis compared to healthy controls. Interestingly, the protein was correlated with psoriasis severity measured by the PASI score. In this sense, we have found similar results in HS patients, although ANGPTL2 was related to the severity of HS independently of BMI, the presence of metabolic syndrome, and serum levels of hs-CRP, indicating that it could be a biomarker of the HS severity by itself.

Our study has limitations inherent to a case–control study. Moreover, as an observational study, it may be subject to some bias due to the possible existence of confounders. However, to try to avoid this issue, adjustment for multiple potential confounding factors has been carried out. The study has been conducted in a single center and has included only Caucasian individuals and, therefore, the results could not be extrapolated to other populations and ethnicities.

## 5. Conclusions

In conclusion, to the best of our knowledge, we have found for the first time, that serum ANGPTL2 concentrations are elevated in HS patients compared to controls and correlate with the duration of the disease. Moreover, ANGPTL2 might serve as a biomarker of HS severity that would indicate an alteration of tissue homeostasis in these patients. Further and larger studies are needed to know the exact role of this glycoprotein in HS pathogenesis and its potential implications in the future therapeutic schemes of this systemic chronic inflammatory disorder.

## Figures and Tables

**Figure 1 biomedicines-11-01204-f001:**
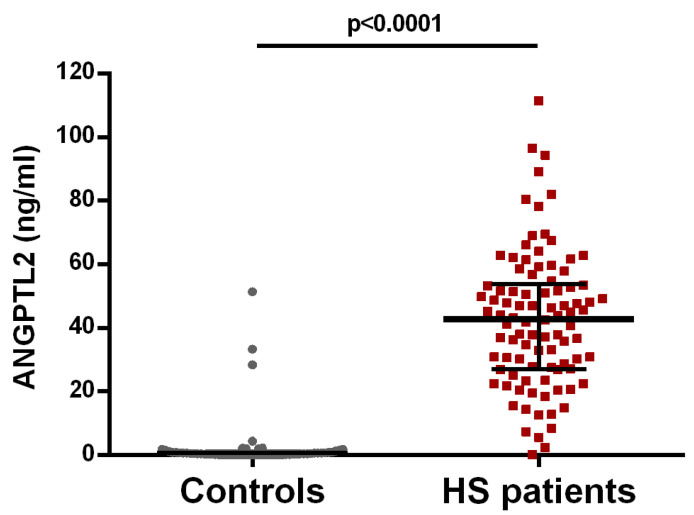
Serum ANGPTL2 levels in HS patients and controls. Median and 25 and 75 interquartile ranges are represented. Significant differences were observed (*p* < 0.0001).

**Figure 2 biomedicines-11-01204-f002:**
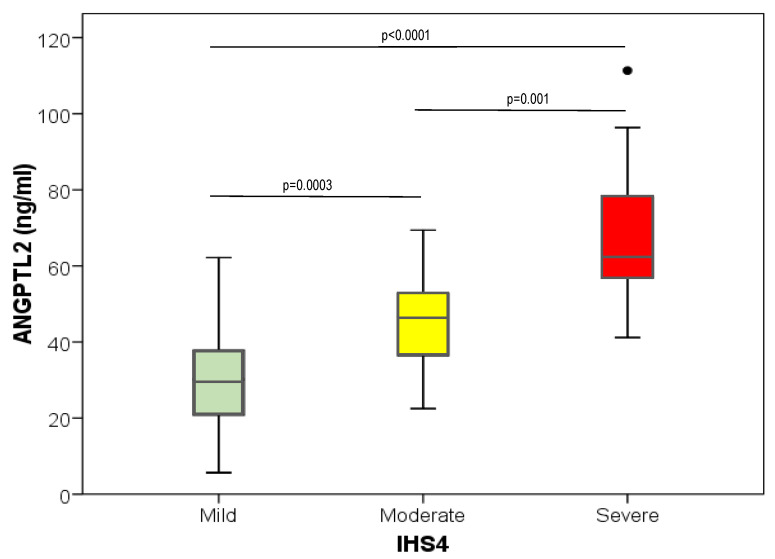
Box and whisker plot representing the serum ANGPTL2 levels in the different HS patients’ groups, according to the HS patients’ severity score (IHS4). Significant differences between groups of patients are shown.

**Figure 3 biomedicines-11-01204-f003:**
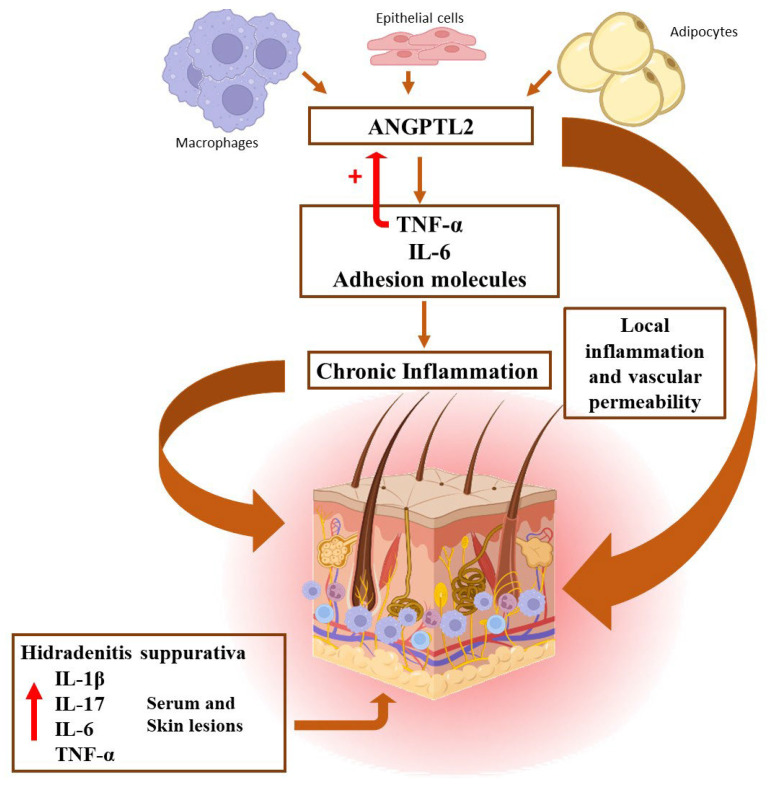
Illustrative scheme showing the pathophysiological mechanisms involved in the development of HS lesions.

**Table 1 biomedicines-11-01204-t001:** Baseline demographic, clinical, and laboratory findings of HS patients and controls.

Variable	HS Patients (N = 94)	Controls (N = 60)	*p*
Age, yrs	42.1 ± 11.4	45.6 ± 12.9	0.07
Sex (female), %	50.0	50.0	1.00
Current smokers, %	70.2	18.3	<0.0001
BMI, kg/m^2^	29.2 ± 5.3	26.5 ± 4.5	0.001
Waist perimeter, cm	99.7 ± 14.4	91.2 ± 13.8	0.001
SBP, mm Hg	130.2 ± 16.3	124.1 ± 15.9	0.025
DBP, mm Hg	80.4 ± 14.1	77.0 ± 8.4	0.09
Metabolic syndrome, %	37.2	11.7	<0.0001
Hypertension, %	16.0	15.0	0.87
Dyslipidemia, %	11.8	16.7	0.39
hs-CRP, mg/dL	0.40 (0.22–0.87)	0.10 (0.10–0.20)	<0.0001
ESR, mm/h	8.0 (3.0–16.5)	13.5 (6.3–24.8)	0.006
HbA1c, %	5.2 ± 0.5	5.2 ± 0.3	0.92
LDL-c, mg/dL	117.0 ± 31.4	122.2 ± 29.4	0.31
HDL-c, mg/dL	46.0 (39.0–55.3)	52.5 (46.3–70.5)	0.001
Triglycerides, mg/dL	101.9 ± 51.8	97.6 ± 67.0	0.65
Fasting plasma glucose, mg/dL	94.8 ± 13.6	89.1 ± 8.1	0.001
Fasting plasma insulin, µIU/mL	9.8 (5.3–16.8)	7.2 (4.8–10.7)	0.01
HOMA-IR	2.20 (1.10–3.75)	1.47 (0.88–2.27)	0.003
IR, %	43.9	20.0	0.003
Fibrinogen, mg/dL	310.0 (268.0–369.0)	266.5 (236.8–298.0)	<0.001
ANGPTL2, ng/mL	45.10 (30.24–57.97)	0.37 (0.09–0.92)	<0.0001

BMI: body mass index; SBP: systolic blood pressure; DBP: diastolic blood pressure; hs-CRP: high-sensitive C-reactive protein; ESR: erythrocyte sedimentation rate; HbA1c: glycated hemoglobin; LDL: low-density lipoprotein; HDL: high-density lipoprotein; HOMA-IR: Homeostatic model assessment for insulin resistance; ANGPTL: angiopoietin-like 2. Values are expressed as percentage mean ± SD or median (interquartile range) as appropriate.

## Data Availability

The data presented in this study are available on request from the corresponding author. The data are not publicly available due to ethical issues.

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
