# Peer review of "Angiopoietin-like 2 Protein and Hidradenitis Suppurativa: A New Biomarker for Disease Severity"

_biomedicines, 2023, doi:10.3390/biomedicines11041204_

Round 1
Reviewer 1 Report
In this Ms by Hernandez et colleagues, the authors aimed to investigate serum ANGPTL2 levels in HS patients and controls and to assess whether ANGPTL2 levels are associated with the severity of HS.
Comments and suggestions:
- A summarized scheme with all the steps included in this study is useful for readers for a better understanding of the MS.
-Why is "metabolic syndrome" mentioned in keywords? Also, these are incomplete.
-Introduction: Lines 46-54: highlight the importance of ANGPTL2 as a biomarker also in other chronic diseases (chronic inflammatory, cardiovascular, metabolic as DM and so on..)
-Figure 1: Legend needs to be expanded with commentary on graphs included, and abbreviations need to be explained. A colour version would be more useful. The same comments for Figure 2.
Discussion section:
Lines 169-186: The mentioned pathophysiological data are difficult to understand and follow, therefore an illustrative scheme with the included mechanisms is needed.
Lines: 188-191: Mention more limitations of this study
What perspectives for human health does this MS have?
Consider revision accordingly.
Author Response
ANSWERS TO REVIEWER 1
First of all, we would like to thank the reviewer for the exhaustive work carried out, providing constructive criticism to our manuscript.
We believe we have responded to his comments and suggestions and have modified the work following them.
We think and hope that the manuscript will now be accepted in its present form.
Reviewer 1
In this Ms by Hernandez et colleagues, the authors aimed to investigate serum ANGPTL2 levels in HS patients and controls and to assess whether ANGPTL2 levels are associated with the severity of HS.
Comments and suggestions:
A1. A summarized scheme with all the steps included in this study is useful for readers for a better understanding of the MS.
Q1. Thank you for your suggestion. We have included a summarized scheme (graphical abstract) for a better understanding of MS following your recommendation.
A2. Why is "metabolic syndrome" mentioned in keywords? Also, these are incomplete.
Q2. We agree with this comment, and, therefore metabolic syndrome was removed and keywords have been completed.
Keywords: Hidradenitis suppurativa; angiopoietin-like 2 protein; insulin-resistance; International Hidradenitis Suppurativa Severity Score System
A3.Introduction: Lines 46-54: highlight the importance of ANGPTL2 as a biomarker also in other chronic diseases (chronic inflammatory, cardiovascular, metabolic as DM, and so on..)
Q3. We have highlighted and expanded this paragraph with some additional information as follows:
Angiopoietin-like-2 protein (ANGPTL2) signaling functions in angiogenesis and tissue repair, whereas an excess of ANGPTL2 signaling has been associated with chronic inflammation and subsequent pathological irreversible tissue remodeling. This protein has been linked to smoking, obesity, and insulin resistance, three key factors in HS pathogenesis [13]. Besides, overexpression of ANGPTL2 in skin tissue leads to focal inflammation and increases blood vessel permeability due to vascular inflammation [14]. Thus, adipocyte-derived inflammatory ANGPTL2 has been proposed to link obesity to insulin resistance.In patients with diabetes and obesity, ANGPTL2 produced by adipocytes, infiltrated macrophages and endothelial cells, leading to an inflammatory response through activation of the integrin α5β1/Rac1NFκB pathway resulting in inflammatory gene expression (Tabata M, Kadomatsu T, Fukuhara S, Miyata K, Ito Y, Endo M, Urano T, Zhu HJ, Tsukano H, Tazume H, Kaikita K, Miyashita K, Iwawaki T, Shimabukuro M, Sakaguchi K, Ito T, Nakagata N, Yamada T, Katagiri H, Kasuga M, Ando Y, Ogawa H, Mochizuki N, Itoh H, Suda T, Oike Y. Angiopoietin-like protein 2 promotes chronic adipose tissue inflammation and obesity-related systemic insulin resistance. Cell Metab. 2009 Sep;10(3):178-88). In the Hisayama study, (Doi Y, Ninomiya T, Hirakawa Y, Takahashi O, Mukai N, Hata J, Iwase M, Kitazono T, Oike Y, Kiyohara Y. Angiopoietin-like protein 2 and risk of type 2 diabetes in a general Japanese population: the Hisayama study. Diabetes Care. 2013;36:98-100. doi: 10.2337/dc12-0166) raised serumANGPTL2 levels were positively associated with the development of type 2 diabetes mellitus in community-dwelling Japanese subjects.
High levels of ANGPTL2 have also been proposed to be a biomarker of chronic inflammatory disorders and early diagnosis, prognosis, and recurrences of some types of cancers. The link between this increase in serum ANGPLT2 levels and these chronic conditions is systemic inflammation and some other properties of this protein indirectly related to inflammation, such as its potential to contribute to cellular senescence.(Thorin-Trescases N, Thorin E. High Circulating Levels of ANGPTL2: Beyond a Clinical Marker of Systemic Inflammation. Oxid Med Cell Longev. 2017;2017:1096385. doi: 10.1155/2017/1096385).
Moreover, since vascular injury along with vascular inflammation is considered an early manifestation of arteriosclerosis, serum ANGPTL2 levels might be involved in the arteriosclerotic process and could act as a new biomarker of atherosclerosis [10]. Thus, ANGPTL2 is abundantly expressed in endothelial cells and macrophages infiltrating atheromatous plaques(Zhang J. Biomarkers of endothelial activation and dysfunction in cardiovascular diseases. Rev Cardiovasc Med. 2022;23:73. doi: 10.31083/j.rcm2302073). In the clinical setting, serum ANGPTL2 levels are highly expressed in patients with acute myocardial infarction and have been positively related to the severity of the coronary lesion (Cao Y, Li R, Zhang F, Guo Z, Tuo S, Li Y. Correlation between angiopoietin-like proteins in inflammatory mediators in peripheral blood and severity of coronary arterial lesion in patients with acute myocardial infarction. ExpTher Med. 2019;17:3495-3500. doi: 10.3892/etm.2019.7386).
A4. Figure 1: Legend needs to be expanded with commentary on graphs included, and abbreviations need to be explained. A colour version would be more useful. The same comments for Figure 2.
Q4. Thank you for your suggestion. We have coloured the graphs and expanded the legends.
Discussion section:
A5. Lines 169-186: The mentioned pathophysiological data are difficult to understand and follow, therefore an illustrative scheme with the included mechanisms is needed.
Q5.Thank you for your comment. We have done the illustrative scheme as you suggest and we have included it in the Ms.
A6. Lines: 188-191: Mention more limitations of this study
Q6. Thank you for this comment. We have added some additional limitations as follows:
Our study has limitations inherent to a case-control study. Moreover, as an observational study, it may be subject to some bias due to the possible existence of confounders. However, to try to avoid this issue, adjustment for multiple potential confounding factors has been carried out. The study has been conducted in a single center and has included only Caucasian individuals and, therefore, the results could not be extrapolated to other populations and ethnicities.
A7. What perspectives for human health does this MS have?
Q7. This is an interesting question. As stated in the “Conclusion” section the main perspective for human health, as reviewers rightly point out is that “ANGPTL2 might serve as a biomarker of H severity that would indicate an alteration of tissue homeostasis in these patients”. It is tempting to speculate that ANGPLT2 inhibitors might play a beneficial role in this chronic systemic disorder, and this has been included in the last paragraph of the “Discussion” section (Further and larger studies are needed to know the exact role of this glycoprotein in HS pathogenesis and its potential implications in the future therapeutic schemes of this systemic chronic inflammatory disorder).

Reviewer 2 Report
This is a clinical study investigating serum concentration of ANGPTL2 in the patients with HS and healthy controls. As a result, the authors found that the mean concentration of ANGPTL2 is significantly high in the HS patients compared with controls. In addition, the authors found that the concentrations of ANGPTL2 well correlate to disease severity of HS. The result was clearly presented, but there raise several points to be clarified or improved in the manuscript.
Major point
1 The authors already published that the serum concentration of angiopoietin-2 was significantly high in the patients with HS compared with controls, and serum concentration of angiopoietin-2 correlates to disease severity of HS (PGA), as presented in the ref. 6. In the Introduction section, the authors should describe why the authors chose ANGPL2 for the severity marker of HS in this study.
2 The authors presented the relation of ANGPTL2 concentration to another disease severity index (IHS4) in this study. The authors should analyze which of the two molecules, angiopoietin-2 and ANGPTL2, indicates better disease severity of HS by using PGA and IHS4, or both indexes.
3 In the Discussion section (page 5, line 147-167), there are long explanation of ANGPL2 function, which should be moved to the Introduction section.
4 In the Discussion section, the authors should explain differences between angiopoietin-2 and ANGPTL2 in the points of functions, secreting cells, and triggers of secretion, and should interpret the result as a biomarker for disease severity of HS.
5 In the Discussion section (page 6, line 169-172), the description was repetitive to the description in the Introduction section (page 2, line 35-41), so that this can be incorporated to Introduction section, or alternatively removed.
6
Minor points
1 Page 2, line 63-64: the diagnostic criteria for HS needs reference.
2 Page 2, line 69-70: the IHS4 needs reference.
3 Abbreviation IR of “insulin resistance” should be consistent through the manuscript. Page 2, line 82, 84, page 3, line 120, Table 1, page 4, line 139, page 5, and line 166.
Author Response
ANSWERS TO REVIEWER 2
First of all, we would like to thank the reviewer for the exhaustive work carried out, providing constructive criticism to the submitted manuscript.
We believe we have responded to his comments and suggestions and have modified the work following them.
We think and hope that the manuscript will now be accepted in its present form.
Reviewer 2
This is a clinical study investigating serum concentration of ANGPTL2 in the patients with HS and healthy controls. As a result, the authors found that the mean concentration of ANGPTL2 is significantly high in the HS patients compared with controls. In addition, the authors found that the concentrations of ANGPTL2 well correlate to disease severity of HS. The result was clearly presented, but there raise several points to be clarified or improved in the manuscript.
Major point
A1. The authors already published that the serum concentration of angiopoietin-2 was significantly high in the patients with HS compared with controls, and serum concentration of angiopoietin-2 correlates to disease severity of HS (PGA), as presented in the ref. 6. In the Introduction section, the authors should describe why the authors chose ANGPL2 for the severity marker of HS in this study.
Q1.Thank you very much for your suggestion. We have explained in more detail why we have chosen ANGPTL2 as a potential biomarker of severity in HS and this information has been added as follows:
In contrast to angiopoietins (Ang), ANGPTLs, despite sharing a great similarity in their amino acid and structural sequences [9], none of these latter bind Tie-1 or Tie-2, the endotelial-specific receptor tyrosine kinase 1 and 2 [20, 21].
Binding of ANGPTL2 to integrin α5β1 enhances cell motility and extracellular matrix remodeling, leading to tissue repair [12]. In the same way, it can bind to type 1A angiotensin II receptors in the cytosol of several cells. Similarly, it binds to CD146, which is present in endothelial cells, preadipocytes, mature adipocytes, and other cells. Furthermore, it binds human leukocyte immunoglobulin-like receptor B2 in bone marrow [22, 23](Deng, M.; Lu, Z.; Zheng, J.; Wan, X.; Chen, X.; Hirayasu, K.; Sun, H.; Lam, Y.; Chen, L.; Wang, Q.; et al. A motif in LILRB2 critical for Angptl2 binding and activation. Blood 2014, 124, 924–935 & Zhuo Y, Wenqian Y, Xiaoxiao H etal.Endothelial cell-derived angiopoietin-like protein 2 supports hematopoietic stem cell activities in bone marrow niches. Blood, 2022; 139: 1529–1540)
Many studies have shown that ANGPTLs play important roles in angiogenesis, lipid metabolism, and inflammation. In particular, ANGPTL2 signaling has been reported to be important for angiogenesis, chronic inflammation, metabolic disease, atherosclerotic diseases, and cancers [9, 10, 13, 16-18].
To our knowledge, the role of serum ANGPTL2 levels in HS has not been assessed to date. In view of the above, and based on our previous work regarding the role of several molecules related to endothelial dysfunction, atherosclerosis, and disease severity in patients with chronic inflammatory disorders, angiopoietin-2 (Ang-2) among others [6], we aimed to investigate whether there could be differences in serum ANGPTL2 levels in HS patients compared with healthy controls. Furthermore, we sought to assess whether there were any relationship between these levels and HS severity.
A2. The authors presented the relation of ANGPTL2 concentration to another disease severity index (IHS4) in this study. The authors should analyze which of the two molecules, angiopoietin-2 and ANGPTL2, indicates better disease severity of HS by using PGA and IHS4, or both indexes.
Q2. This study is devoted to ANGPTL2 and we have data focused on this protein. We previously published a manuscript regarding the role of angiopoietin-2 and other molecules in HS. The comparison between bot ANGPT2 and Ang-2 was not performed because this issue was not the purpose of this work. However, we agree that this is an interesting point which could be the subject of another study.
A3.In the Discussion section (page 5, line 147-167), there are long explanation of ANGPL2 function, which should be moved to the Introduction section.
Q3. Thank you for your accurate comment. Following your suggestion, these paragraph has been moved to the “Introduction section”
ANGPTL2 is a glycoprotein belonging to the angiopoietin-like (ANGPTL) family, with an N-terminal coiled-coil domain, a short linker peptide, and a C-terminal fibrinogen-like domain [9]. It is highly expressed in adipose tissues and may play a pivotal role in some inflammatory processes such as obesity-related IR, chronic systemic inflammatory diseases such as rheumatoid arthritis or dermatomyositis [10-13].
A4. In the Discussion section, the authors should explain differences between angiopoietin-2 and ANGPTL2 in the points of functions, secreting cells, and triggers of secretion, and should interpret the result as a biomarker for disease severity of HS.
Q4. Thank you very much for your comment. As you suggest, we explain differences between Ang-2 and ANGPTL2 and include them in the Discussion section.
Angiopoietin-2 is a multifaceted protein expressed by vascular endothelium that acts as an antagonist of Angiopoietin-1 and the Tie2 receptor.(Scholz A, Plate KH, Reiss Y. Angiopoietin-2:a multifaceted cytokine that functions in both angiogenesis and inflammation. Ann N Y Acad Sci. 2015;1347:45–51).
The Ang/Tie pathway acts as a regulator of angiogenesis under physiological and pathological conditions [Wang L, Chen Q, Pang J. The effects and mechanisms of ghrelin upon angiogenesis in human coronary artery endotelial cells under hipoxia. Peptides 2023 Feb;160:170921. doi: 10.1016/j.peptides.2022.170921]..Ang-1 and Ang-2 bind to the endotelial-specific receptor tyrosine kinase (Tie2) with similiar affinities but with different effect. Ang-1 induces Tie2 phosphorylation while Ang-2 which is the natural antagonist of Ang-1, blocks Tie2 phosphorylation [20]
Ang-2 is involved in angiogenesis, endotelial activation, atherosclerotic processes, and inflammatory responses [29]. As has been observed in other chronic inflammatory diseases, we previously reported that serum Ang-2 levels were significantly higher in patients with HS compared to controls [6]. Furthermore, Ang-2 was positively correlated with disease severity in HS patients. In this regard, a positive correlation between serum Ang-2 levels and disease severity has also been found in other dermatological diseases such as psoriasis [11] and atopic dermatitis [12]. Since th ecorrelation between Ang-2 and CV disease is well known, the results of our study might indicate that high levels of Ang-2 could play a role in the CV burden of HS.
In contrast to angiopoietins, ANGPTLs do not bind Tie-1 or Tie-2. Many studies have shown that ANGPTLs play important roles in angiogenesis, lipid metabolism, and inflammation. In particular, ANGPTL2 signaling has been reported to be important for angiogenesis, chronic inflammation, metabolic disease, atherosclerotic diseases, and cancers.
Moreover, ANGPTL2 binds to integrin α5β1 to enhance cell motility and extracellular matrix remodeling, leading to tissue repair. In the same way, it can bind to type 1A angiotensin II receptors in the cytosol of several cells. Similarly, it binds to CD146, which is present in endothelial cells, preadipocytes, mature adipocytes, and other cells. Furthermore, it binds human leukocyte immunoglobulin-like receptor B2 in bone marrow (Endothelial cell-derived angiopoietin-like protein 2 supports hematopoietic stem cell activities in bone marrow niches. Blood, 2022; 139: 1529–1540).
A5.In the Discussion section (page 6, line 169-172), the description was repetitive to the description in the Introduction section (page 2, line 35-41), so that this can be incorporated to Introduction section, or alternatively removed.
Q5. Thank you for your comment. Nevertheless, we think that this paragraph is important to build the discussion of the manuscript. It is crucial to highlight the HS pathogenesis related to the effect of proinflammatory cytokines in this disorder.
Minor points
A1. Page 2, line 63-64: the diagnostic criteria for HS needs reference.
Q1.Thank you for the appreciation. Two references regarding HS diagnostic criteria has been added.
A2. Page 2, line 69-70: the IHS4 needs reference.
Q2.Thank you for the appreciation. The reference of this dynamic scoring system to assess the severity of HS has been added.
A3. Abbreviation IR of “insulin resistance” should be consistent through the manuscript. Page 2, line 82, 84, page 3, line 120, Table 1, page 4, line 139, page 5, and line 166.
Q3. Thank you for your suggestion. We have modified it

Reviewer 3 Report
Dear Authors,
An interesting manuscript, but it rises some questions/remarks besides the originality:
1) Introduction. Please, discuss also other possible cases/diseases/reasons for ANGPTL-2 changes in the serum. This info fits here and the reader feels the curiosity about this.
2) Material and methods. Please, clarify the inclusion/exclusion criteria for the controls. Also give some info about the gender and reason why exact age patients were chosen. Well, topical is also forever question how you can be sure about the lack of other side factors/confounding factors that may influence the result.
3) Discussion. Please, include some additional points: - very rare just some one factor works as a diagnostic marker. Thus, please add some speculation on the possible influence of other factors on the ANGPTL-2 level; also add some ideas about the combination of ANGPTL-2 and some other factors (cytokines, some other indices) for the pathogenesis of disease.
4) Conclusions. remove, please, the first part of the 2st sentence, this is extra and useless.
5) References. Sorry, highly no enough. Add info in the Introduction and Discussion, this will help increase the number of literature sources.
Author Response
ANSWERS TO REVIEWER 3
First of all, we would like to thank the reviewer for the exhaustive work carried out, providing constructive criticism to the submitted manuscript.
We believe we have responded to his comments and suggestions and have modified the work following them.
We think and hope that the manuscript will now be accepted in its present form.
Reviewer 3
Dear Authors,
An interesting manuscript, but it rises some questions/remarks besides the originality:
A1. Introduction. Please, discuss also other possible cases/diseases/reasons for ANGPTL-2 changes in the serum. This info fits here and the reader feels the curiosity about this.
Q1.Thank you for your accurate comment. This information has been added to the Introduction section:
Angiopoietin-like-2 protein (ANGPTL2) signaling functions in angiogenesis and tissue repair, whereas an excess of ANGPTL2 signaling has been associated with chronic inflammation and subsequent pathological irreversible tissue remodeling. This protein has been linked to smoking, obesity, and insulin resistance, three key factors in HS pathogenesis [13]. Besides, overexpression of ANGPTL2 in skin tissue leads to focal inflammation and increases blood vessel permeability due to vascular inflammation [14]. Thus, adipocyte-derived inflammatory ANGPTL2 has been proposed to link obesity to insulin resistance.In patients with diabetes and obesity, ANGPTL2 produced by adipocytes, infiltrated macrophages and endothelial cells, leading to an inflammatory response through activation of the integrin α5β1/Rac1NFκB pathway resulting in inflammatory gene expression (Tabata M, Kadomatsu T, Fukuhara S, Miyata K, Ito Y, Endo M, Urano T, Zhu HJ, Tsukano H, Tazume H, Kaikita K, Miyashita K, Iwawaki T, Shimabukuro M, Sakaguchi K, Ito T, Nakagata N, Yamada T, Katagiri H, Kasuga M, Ando Y, Ogawa H, Mochizuki N, Itoh H, Suda T, Oike Y. Angiopoietin-like protein 2 promotes chronic adipose tissue inflammation and obesity-related systemic insulin resistance. Cell Metab. 2009 Sep;10(3):178-88). In the Hisayama study, (Doi Y, Ninomiya T, Hirakawa Y, Takahashi O, Mukai N, Hata J, Iwase M, Kitazono T, Oike Y, Kiyohara Y. Angiopoietin-like protein 2 and risk of type 2 diabetes in a general Japanese population: the Hisayama study. Diabetes Care. 2013;36:98-100. doi: 10.2337/dc12-0166) raised serumANGPTL2 levels were positively associated with the development of type 2 diabetes mellitus in community-dwelling Japanese subjects.
High levels of ANGPTL2 have also been proposed to be a biomarker of chronic inflammatory disorders and early diagnosis, prognosis, and recurrences of some types of cancers. The link between this increase in serum ANGPLT2 levels and these chronic conditions is systemic inflammation and some other properties of this protein indirectly related to inflammation, such as its potential to contribute to cellular senescence.(Thorin-Trescases N, Thorin E. High Circulating Levels of ANGPTL2: Beyond a Clinical Marker of Systemic Inflammation. Oxid Med Cell Longev. 2017;2017:1096385. doi: 10.1155/2017/1096385).
Moreover, since vascular injury along with vascular inflammation is considered an early manifestation of arteriosclerosis, serum ANGPTL2 levels might be involved in the arteriosclerotic process and could act as a new biomarker of atherosclerosis [10]. Thus, ANGPTL2 is abundantly expressed in endothelial cells and macrophages infiltrating atheromatous plaques(Zhang J. Biomarkers of endothelial activation and dysfunction in cardiovascular diseases. Rev Cardiovasc Med. 2022;23:73. doi: 10.31083/j.rcm2302073). In the clinical setting, serum ANGPTL2 levels are highly expressed in patients with acute myocardial infarction and have been positively related to the severity of the coronary lesion (Cao Y, Li R, Zhang F, Guo Z, Tuo S, Li Y. Correlation between angiopoietin-like proteins in inflammatory mediators in peripheral blood and severity of coronary arterial lesion in patients with acute myocardial infarction. ExpTher Med. 2019;17:3495-3500. doi: 10.3892/etm.2019.7386).
A2. Material and methods. Please, clarify the inclusion/exclusion criteria for the controls. Also givesome info about the gender and reason why exact age patients were chosen. Well, topical is also forever question how you can be sure about the lack of other side factors/confounding factors that may influence the result.
Q2. Thank you for your comment. We have clarified these criteria.Patients or controls with personal history of CV events, cancer, diabetes mellitus, chronic kidney or liver failure, and/or other inflammatory diseases were dicarded in order to avoid posible influence of inflammatory processes which could increase ANGPTL2 levels.
By the other hand, ANGPTL2 levels can be influenced by age or sex, and that was one of the reasons to choose a case-control design with controls of similar age and sex. We have adjusted for a wide set of confounders already known to potentially influence the results, as the reviewer can observe in the statistical analysis section. Multivariable models were built to try to control for the main potential confounding factors.
A3. Discussion. Please, include some additional points: - very rare just some one factor works as a diagnostic marker. Thus, please add some speculation on the possible influence of other factors on the ANGPTL-2 level; also add some ideas about the combination of ANGPTL-2 and some other factors (cytokines, some other indices) for the pathogenesis of disease.
Q3.Thank you very much por your comment.
We agree with you that ANGPTL2 alone would not be very useful as a biomarker of HS or HS severity, but the novelty of our work, like that of previous works with other biomarkers, is that in this work, we have described, for the first time, the association of this protein both with HS and with its severity degree. Probably the use of ANGPTL2, together with other previously reported biomarkers would help in the diagnosis and prognosis of HS.
Regarding your suggestion to add some ideas about the combination of ANGPTL2 and some other factors, we have include some comments in the Discussion section:
HS is now recognized as a systemic disease. The precise inflammatory mechanism underlying HS pathogenesis is still not fully elucidated, but elevated levels of proinflam-matory cytokines, such as TNFα, IL-6, IL-17, and IL-1 have been observed in serum and HS skin lesions [2, 3, 30, 31]. In this sense, ANGPTL2 may induce the expression of proin-flammatory cytokines, such as TNFα, IL-6, and some adhesion molecules, that have been involved in the pathogenesis of HS and may contribute to maintaining a chronic inflam-matory status characteristic of this disorder. Besides, TNFα regulates ANGPTL2 mRNA expression and could potentiate the overall inflammatory effect [32]. In mice, it has been reported that ANGPTL2 exacerbates proinflammatory signaling by activating the Integrin α5β1/NF-κB pathway in coronary endothelial cells [26]. The nuclear factor-κB (NF-κB) is a transcription factor which develops a pivotal role in activating both innate and adaptive immunity, mainly leading to target cells to increase production of proinflammatory cyto-kines [33], such as IL-1, TNFα and IL-6. The activation via receptors such as integrin α5β1 and (LILRB2) would trigger a transduction signaling cascade increasing expression of inflammatory factors genes through NF-κB and other transcription factors, as has been demonstrated in synovial tissue [34]. NF-κB pathway activation has been clearly associated with several inflammatory diseases, such as such as rheumatoid arthritis, inflam-matory bowel disease, multiple sclerosis and asthma [33].
A4. Conclusions. remove, please, the first part of the 2st sentence, this is extra and useless.
Q4.Thank you for your comment. Following your suggestion, we have removed it.
A5. References. Sorry, highly no enough. Add info in the Introduction and Discussion, this will help increase the number of literature sources.
Q5. Thank you for your comment. Former references have been reordered and new ones added as the reviewer suggested.

Round 2
Reviewer 1 Report
No answer given.
Author Response
We greatly appreciate your effort into reviewing our work. Your comments and suggestions have helped to enrich and increase the quality of our work.
Reviewer 2 Report
HS is relatively rare dermatosis, of which pathophysiology is not well understood. The results of this study is beneficial to develop its new treatment options. The revised version of manuscript is corrected adequately in the most points given in the comments. However, the version needs additional revision. In the Discussion section, page 6, line 193-229: the description explaining ANGPTL2 and Ang-2 is rather boring. The authors should discuss and rewrite the beneficial points to focus on this study for ANGPTL2 instead of Ang-2.
Author Response
Dear reviewer,
We greatly appreciate your effort into reviewing our work. Your comments and suggestions have helped to enrich and increase the quality of our work.
We have modified some details of the discussion to try to make it less boring. Perhaps the fact that we have to lengthen the manuscript by more than 1,000 words, and that the other two reviewers asked us for more information regarding both molecules has been the cause of this feeling.
Regarding your suggestion that we should discuss and rewrite the beneficial points to focus on this study for ANGPTL2 instead of Ang-2, we just want to tell you that in a recent work previously published by our group in a Q1 journal, we have already addressed the role of Ang-2 in HS (Reference 6. González-López MA, Ocejo-Viñals JG, López-Sundh AE, et al. Biomarkers of endothelial dysfunction and atherosclerosis in hidradenitis suppurativa. J Dermatol. 2022; 49(10): 1052- 6. 2022/06/07. doi: 10.1111/1346-8138.16484).
We hope that our answers are to your liking and you accept the explanations given.
Sincerely
Reviewer 3 Report
Dear Authors,
thank you for the done job. I will advice to publish your manuscript.
Author Response
We greatly appreciate your effort into reviewing our work. Your comments and suggestions have helped to enrich and increase the quality of our work.
Sincerely